# Multifunctional human visual pathway-replicated hardware based on 2D materials

Zhuiri Peng[1], Lei Tong[2], Wenhao Shi [1], Langlang Xu[1], Xinyu Huang[1], Zheng Li[1], Xiangxiang Yu[1], Xiaohan Meng[1], Xiao He[1], Shengjie Lv[1], Gaochen Yang[1], Hao Hao[3], Tian Jiang [3] ✉, Xiangshui Miao [1,4] ✉ & Lei Ye [1,4] ✉

Artificial visual system empowered by 2D materials-based hardware simulates the functionalities of the human visual system, leading the forefront of artificial intelligence vision. However, retina-mimicked hardware that has not yet fully emulated the neural circuits of visual pathways is restricted from realizing more complex and special functions. In this work, we proposed a human visual pathway-replicated hardware that consists of crossbar arrays with split floating gate 2D tungsten diselenide ($WSe_2$) unit devices that simulate the retina and visual cortex, and related connective peripheral circuits that replicate connectomics between the retina and visual cortex. This hardware experimentally displays advanced multi-functions of red–green color-blindness processing, low-power shape recognition, and self-driven motion tracking, promoting the development of machine vision, driverless technology, brain–computer interfaces, and intelligent robotics.

An artificial visual system empowered by hardware aims to replicate the functionalities of the human visual system; its advanced performance in perceiving and processing external visual information is the cornerstone of various fields, such as driverless technology, brain–computer interfaces, and intelligent robots[1–10]. To achieve powerful human visual abilities, artificial visual hardware is developed quickly on the basis of new optoelectronic materials[11–14] and non-von Neumann architectures[15–18]. 2D materials are good candidates[19,20] for fabricating human visual hardware due to their inherently dangling-bond-free surfaces, atomically sharp interfaces, strong light–matter interaction, and electrically tunable photoresponse. For example, by introducing nonvolatile storage, such as ferroelectric[21], floating gate[22,23], and material defect[24], 2D materials have been proven to exhibit reconfigurable optical responsivity to mix in situ sensing preprocessing, edge computing, and signal coding functionalities in one device[25], which is designed as the cornerstone for mimicking the retina. This retina-mimicking design realizes basic functions of human visual adaptation[24,26], color perception[27], feature extraction[22,28–30], and motion sensing[23,31–33]. However, most forms of hardware disregard the

replication of visual pathways in hardware design, making combining all basic functionalities in one hardware to enable more complex and efficient functions a challenging task.

The human visual system is dominated by visual pathways, which include the retina, visual cortex, and connectomics between them (Fig. 1a)[34–36]. In the retina, photoreceptor (rod and cone), bipolar, and ganglion cells are connected successively, horizontal and amacrine cells act on neighbor pixel cells, constituting a center-surround receptive field (CSRF) with potentiated center and depressed surround. The lateral geniculate nucleus (LGN) in the pulvinar is regarded as a connector between the retina and visual cortex to receive the same CSRF information from the parasol and midget ganglion cells in the retina and then send them to the visual cortex for distributed hierarchical processing. The visual cortex consists of the primary visual cortex (V1), secondary visual cortex (V2), area V4 and inferotemporal (IT) cortex in the ventral stream, and middle temporal (MT) cortex and parietal cortex in the dorsal stream. Following the anatomical structure and connectivity of the three aforementioned organizational modules, the P and M pathways[37,38] are constructed to process static

[1]School of Integrated Circuits, Wuhan National Laboratory for Optoelectronics, Huazhong University of Science and Technology, Wuhan, China. [2]Department of Electronic Engineering, Materials Science and Technology Research Center, The Chinese University of Hong Kong, Hong Kong, China. [3]College of Advanced Interdisciplinary Studies, National University of Defense Technology, Changsha, China. [4]Hubei Yangtze Memory Laboratories, Wuhan, China. ✉e-mail: tjiang@nudt.edu.cn; miaoxs@hust.edu.cn; leiye@hust.edu.cn

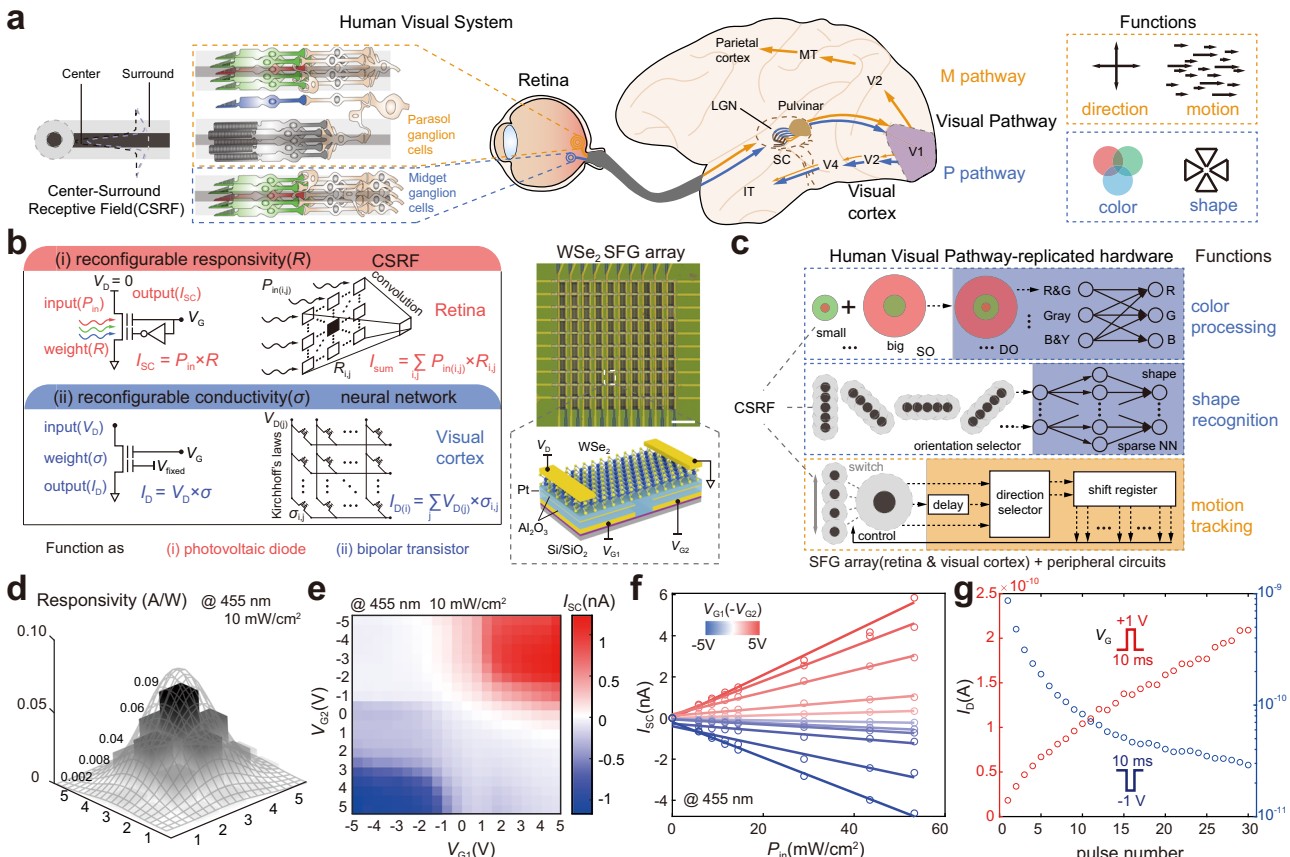

**Fig. 1 | Human visual pathways and working mechanisms of the human visual pathway-replicated hardware. a** Human visual pathways, characteristics of the center-surround receptive field (CSRF), and functions of the human visual system. The schematic diagram is adapted with permission from refs. 35,36. SC superior colliculus, LGN lateral geniculate nucleus, V1 primary visual cortex, V2 secondary visual cortex, V4 extrastriate cortex, MT middle temporal cortex, IT inferotemporal cortex. **b** Microscope image of a 10×10 tungsten diselenide (WSe₂) split floating gate (SFG) diode/transistor array mimicking the retina and visual cortex. Insert: Schematic of a unit device, exhibiting reconfigurable responsivity and conductivity under the photovoltaic diode and bipolar transistor modes. The array under the photovoltaic diode and bipolar transistor modes performs photoresponse

convolution and matrix multiplication to mimic the retina and visual cortex, respectively. **c** Flowchart of the human visual pathway-replicated hardware that consists of the SFG array and connective peripheral circuits for color process, shape recognition, and motion tracking. The shadings in the background represent the parts of peripheral circuits. SO single color opponency receptive field, DO double color opponency receptive field. **d** Optical responsivity distribution of the 5×5 device array. **e** $I_{SC} - V_{G1}$ and $V_{G2}$ mapping under an illumination of 455 nm and 10 mW/cm², showing reconfigurable photoresponse. **f**, $I_{SC}-P_{in}$ curves under different $V_{G1}$ ($-V_{G2}$). **g** Voltage pulse number modulated $I_D$, demonstrating synapse-like potentiation and depression behavior. The pulse is ±1 V/10 ms ($V_{D-read}$ = 0.1 V).

[color[39,40] and shape[41,42]] and dynamic [direction[43,44] and motion[45,46]] information, respectively, beyond the ability of the retina.

To develop an artificial visual system that supports multi-complex functions with lower power consumption, hardware that resembles the visual pathways of the human visual system is under debate. In this work, we designed a general hardware architecture with the crossbar array and related connective peripheral circuits to replicate the neural circuits of visual pathways. The basic device in the crossbar array is a tungsten diselenide (WSe₂) split floating gate (SFG) device with reconfigurable positive/negative optical responsivity and conductivity, enabling the crossbar array to simulate the CSRF of the retina and neural networks of the visual cortex in the human visual system. The connection between the retina and cortex is based on the related peripheral circuits. The SFG arrays are used to construct related visual pathway-replicated hardware with specific peripheral circuits, achieving color vision, shape vision, and dynamic vision. The retina-like operation of the array under photovoltaic diode mode is self-powered with nearly zero standby power consumption, and the cortex-like operation under bipolar transistor mode exhibits the programming energy for a floating gate of below 1 pJ/spike, promising ultralow power consumption. On the basis of the visual pathway-replicated

design, the hardware experimentally performs color processing in accordance with the human visual system, making it capable of explaining the cause of red–green color blindness (Daltonism) with the hardware. The shape vision hardware is also demonstrated through an effective shape classification within a double-layer sparse neural network, demonstrating neural circuit-compatible sparsity and a recognition rate of >95% in the experiment. The promise of low-power applications is confirmed by a 61.1% reduction in device usage and only 0.9 nJ programming energy per operation. The hardware realizes human dynamic vision function to track motion information in real-time by processing the transmission time difference of synapses within the visual pathway. Remarkably, the diverse functions presented in the human visual pathway-replicated hardware prove that it is a powerful platform for artificial intelligence visual tasks.

## Results

### Working mechanisms of hardware

As the core building block, the 10×10 crossbar array is fabricated with a unit SFG device of Al₂O₃/Pt/Al₂O₃/WSe₂, which functions as a photovoltaic diode or a bipolar transistor (Fig. 1b). Both working modes are required to build the visual pathway in one hardware. For

photovoltaic diode mode under two modulated opposite SFGs, the nonvolatile reconfigurable n–p junction results in photovoltaic effect-based positive and negative photocurrents with adjustable optical responsivity to construct CSRF by convoluting input light with a responsivity kernel. For bipolar transistor mode under one gate with fixed $V_G$, the reconfigurable conductivity serves as the unit weight of the neural network to realize the visual cortex that can calculate matrix multiplication by Kirchhoff's laws. To replicate the visual pathways of the human visual system in the hardware, $WSe_2$ SFG arrays are successively integrated to mimic the retina or visual cortex by building related peripheral circuits for current-to-voltage transition and gate voltage programming and control (Fig. 1c). Then, the multifunctions of color processing, shape recognition, and motion tracking are realized. In particular, the retina-related parts of the hardware, including color sensing, orientation selector, and motion tracking, are built with arrays that work in photovoltaic diode mode. Meanwhile, visual cortex-related parts, including the CSRF of single color opponency (SO) and double color opponency (DO), neural network for color processing, and double-layer sparse neural network, are built with arrays that work in bipolar transistor mode. These functions require exquisite fitting of visual pathways, which are supported by the designed hardware, as discussed in the following sections.

Under photovoltaic diode mode, the reconfigurable optical responsivity modulated by gates $V_{G1}$ and $V_{G2}$ is the basis for realizing CSRF. When $V_{G1}$ and $V_{G2}$ have the same values and opposite signs, the $I_D$–$V_D$ curves (Fig. S4b) under the illumination of red–green–blue light (637 nm/520 nm/455 nm, $P_{in} = 10$ mW/cm$^2$) exhibit typical p–n junction photovoltaic effect. The quasi-linear adjustment of positive/negative optical responsivity can be achieved by gate voltages (Fig. S4c), making reconstructing CSRF an easy task. Here, the impact of non-linear dependency on subsequent imaging processing is suppressed by fixing the weights within several discrete values, also as demonstrated in the recent work[25]. The near-Gaussian distributed optical responsivity matrix of a $5 \times 5$ array is constructed under 455 nm and 10 mW/cm$^2$ illumination, where each pixel is independently modulated by gates (Fig. 1d). The SFG voltages significantly co-modulate the unit device (Fig. 1e, S5c, S5d) to construct different junction states (Fig. S3b–S3d). The short-circuit current ($I_{SC}$) of the SFG device is linearly correlated with light intensity ($P_{in}$) (Fig. 1f, S5a, S5b), contributing to the stable coding of incident light information. Furthermore, we test the electronic and optoelectronic performance of each unit in the $10 \times 10$ array and extract performance features (Supplementary Note 1, Figs. S6–S11), exhibiting considerable uniformity, remarkable storage capacity, and excellent gate control for the whole units. Under bipolar transistor mode, gate pulse modulated output updates (Fig. 1g and S8) are measured by $\pm 1$ V/10 ms voltage pulses on one floating gate, resulting in nonvolatile modulation with programming energy below 1 pJ/spike (Supplementary Note 1). The adjustable conductance acts as the unit weight to mimic visual cortex computations in the crossbar array. Thus, each unit of the hardware works properly to meet the functional requirements of the current application.

## Color processing

The biomechanism of color processing in the P pathway is shown in Fig. 2a. Three types of cones are individually sensitive to red/green/blue light, sequentially connected to ON/OFF bipolar cells and ON/OFF ganglion cells. Accepting center-potentiated information from one color cone and surround-depressed information from another one, red–green (R–G/G–R), yellow–blue [(G + R)–B/B–(G + R)], and black–white [(B + G + R)/–(B + G + R)] SO CSRFs in LGN are constructed. In V1, red–green (R&G), blue–yellow (B&Y), and black–white (gray) DO CSRFs are formed by integrating small and large SO CSRF from LGN. Finally, color information is obtained in V4/IT through neural connection analysis from DO CSRF data (details in Supplementary Note 2).

The color processing hardware (Fig. S12) is constructed by SFG arrays and replicates the neural circuit as mentioned above. Mimicking the retina, one SFG array is reused 12 times in the front working under photovoltaic diode mode. The four other arrays that work in bipolar transistor mode replicate the visual cortex LGN–V1–V4–IT successively to obtain voltage signals that represent SO/DO CSRF and color information sequentially through matrix multiplication (Supplementary Note 2).

To demonstrate the working process of the hardware, a $34 \times 31$-pixel trichromatic circular image is used as the light input, decomposed by RGB components, and mapped to light intensity ($P_{in}$) with a wavelength of 637/520/455 nm (Fig. 2a). The light input illuminates the SFG array pixel by pixel (Fig. S1) to perform convolution with color-related kernels. The optical responsivity distribution of color-related kernels for the SO CSRF is set strictly in accordance with the Gaussian function, similar to the biological mechanisms of retina–LGN. The Gaussian standard deviation $\sigma$ determines the degree of blur in image processing, a larger one is fuzzier to extract global information, and a small one is more concentrated to get local information. The $\sigma = 0.08$ and $\sigma = 0.12$ respectively set the responsivity distribution of small (3×3) and large (5×5) CSRF with two values (0.04 and −0.008 A/W) and three values (0.007, 0.003 and −0.0001 A/W) (Fig. 2b and other circumstances in Fig. S13). The photocurrents of the SFG array are added and converted into photovoltages by using a transimpedance amplifier (TIA) in the peripheral circuit. The photovoltage signals of the large and small CSRF are imported into another SFG array to obtain the R + G − /brightness/Y + B− and G + R − /darkness/B + Y − SO CSRF signals through matrix multiplication (Fig. 2c). Among them, "R/G/B/Y" is the color information of red, green, blue, and yellow. The front, back, "+", "−", "lightness", and "darkness" represents center, surround, potentiation, depression, "(B + G + R)", and "−(B + G + R)" respectively. The R&G/gray/B&Y DO CSRF signals and processed RGB components are subsequently obtained through the same process of matrix multiplication. The simulation and experiment results of the DO CSRF and processed RGB components are presented in Fig. 2d, e, respectively. The processed RGB components share the same characteristics of the color information processing pathway of the human visual system, which is hardly realized in traditional color cameras that merely filter RGB components. Moreover, this hardware can demonstrate the unexplained causes of red–green color blindness. Daltonism originates from the failure of the R–G SO–DO pathway[39,40], and thus, only a 10% ratio of weights is applied to this pathway in the experiment setup (×0.1, in the upper plane of Fig. 2c, d). Specifically, the conductance value of the $WSe_2$ transistor connecting the R–G SO and DO is set to 100 nS, while the other channels are set to 1 μS. After the above perception and post-processing procedure, the output color information mimics the perception of color images by Daltonism patients (Fig. 2f), which can only be elucidated by considering the visual pathway.

## Shape recognition

The function of shape recognition is also executed in the P pathway (Fig. 3a). CSRFs in the retina–LGN are integrated into the orientation-selective CSRF in V1 in accordance with different spatial distributions. The contour information of each point is summarized through V2/V4 with a sparse connection, realizing shape classification in IT[47,48].

By replicating the above neural circuit, the shape recognition hardware (Fig. S14) is also constructed by the SFG arrays. One array working in photovoltaic diode mode adopts the photoresponsivity configuration of the orientation convolution kernel (OCK) and is reused to collect the photovoltage of five points, mimicking the retina–LGN. Similar to the visual cortex V1–V2–V4–IT, three other arrays that are working in bipolar transistor mode build a double-layer sparse neural network for shape recognition. This neural network

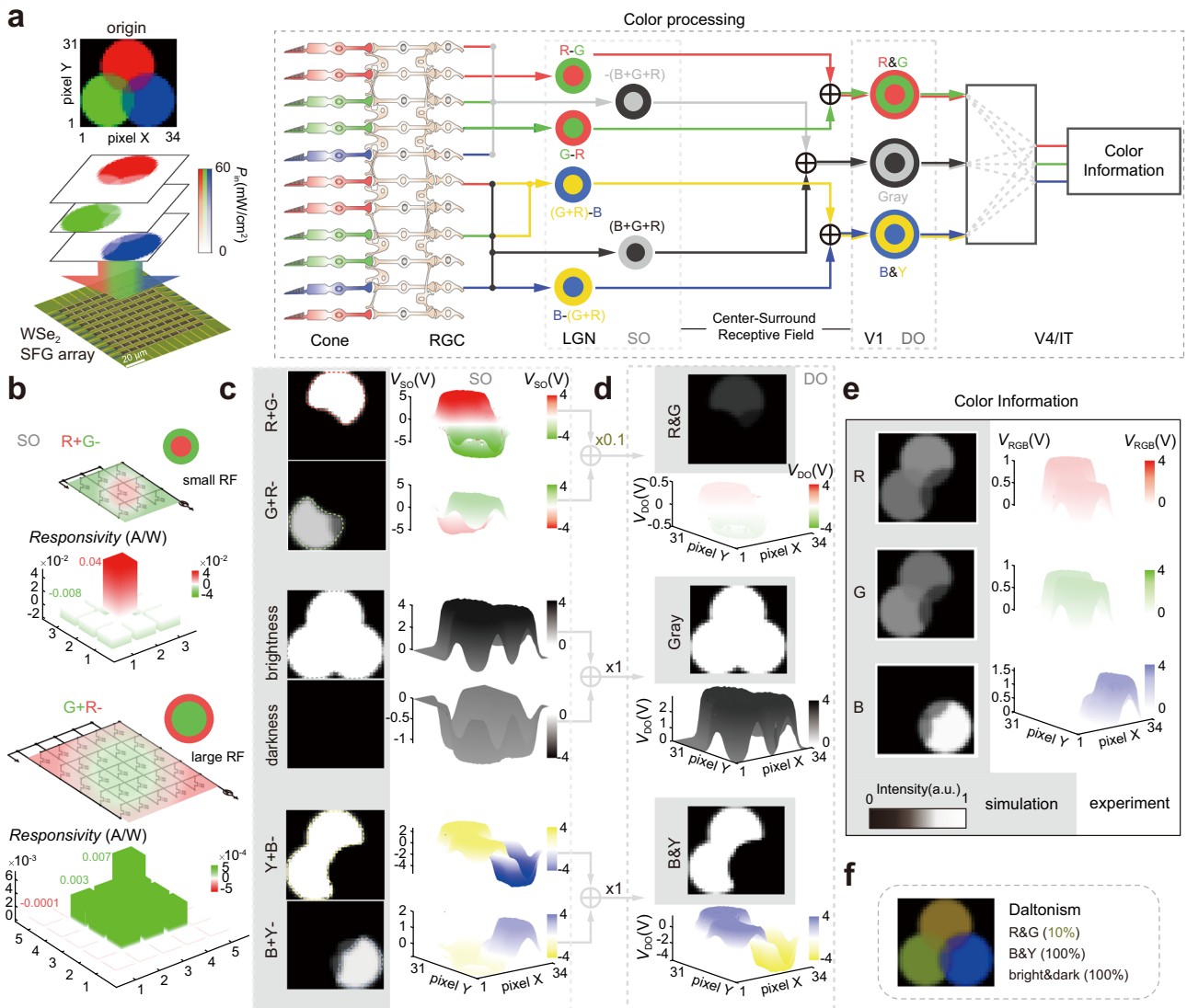

**Fig. 2 | Color processing of Daltonism. a** Light input of a 34 × 31-pixel trichromatic circular image with RGB components onto the tungsten diselenide (WSe₂) split floating gate (SFG) array. Color vision in the human visual system. The single color opponency (SO) receptive field in the lateral geniculate nucleus (LGN) receives center-potentiated and surround-depressed signals of different colors from the retinal ganglion cells (RGC) in the retina. The double color opponency (DO) receptive field in the primary visual cortex (V1) integrates large and small SO signals in the same color type. They are both center-surround receptive fields (CSRFs). In area V4 and inferotemporal (IT) cortex, color information is analyzed by the neural network. **b** Hardware photoresponsivity distributions that correspond to red−green SO CSRF (small CSRF & large CSRF). Simulation (2D gray image) and experiment (3D voltage mapping) results of SO CSRF (**c**), DO CSRF (**d**), processed color information (**e**). Color bar: pixel intensity by simulation, voltage amplitude by experimental test with different individual color bars in each result. The gray shadings mark the results and color bar in simulation. **f** Test result of red−green color blindness (Daltonism) with 10% R&G pathway contribution (the weight coefficient of R&G integration from **c**−**d** is 0.1).

performs matrix multiplication of the 1st layer, activation, and 2nd layer successively (Supplementary Note 3).

The CSRF OCK of the orientation selector is configured by the photoresponsivity distributions of the SFG array, which exhibits positive photoresponsivity along the selective orientation and negative photoresponsivity in other points (Fig. 3b). To test OCK's orientation selection effect, a regular hexadecagon of 50 × 50 pixels is used as the incident light pattern and convolved with different OCKs pixel by pixel to obtain the short-circuit photocurrent distributions. The peak value in the corresponding direction encoded by the OCK is produced as verification. In accordance with the abstract processing flow of the visual pathway, shape recognition is verified in the simulation and experiment (Fig. 3c). The light illuminates the light mask of a right-angled triangle (Fig. S1), and the transmitted light pattern is quantified to the 15 × 15−pixel size as the light-mask input. Using 1600

samples augmented by adding Gaussian noise (with a standard deviation of 0.8) as the database (Fig. S15a), five regions (5 × 5−pixel size for each region) of each sample are divided for convolution with OCKs. In the experiment, the light input state of these five regions is adjusted by moving the position of the mask (Fig. S1). The results undergo feedforward calculation, activation (Fig. S15f), and back-propagation to update the weights through a 5 × 8 × 4 double-layer sparse neural network. Simulation (Fig. 3d) and experiment (Fig. 3e) weights are recorded for 30 epochs. Hardware conductance weight accuracy is quantized into 64 levels (Fig. S15b). After training, the recognition rates of the triangles obtained by simulation and experiment are higher than 95% (Fig. 3f). Although the sparse neural network inspired by the sparsity of neural connections exhibits slight recognition accuracy decay compared with the fully connected network, it considerably reduces device usage and programming energy by

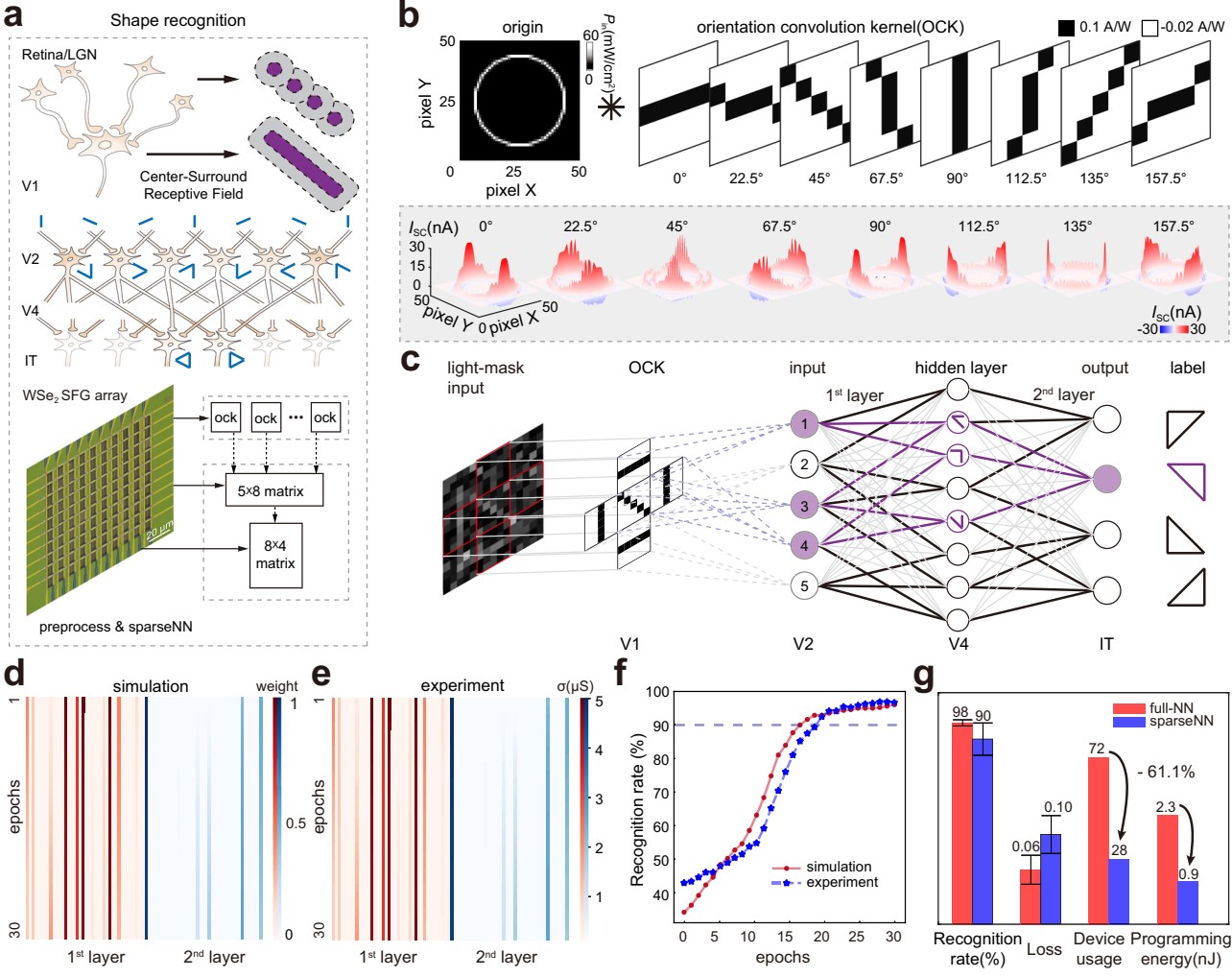

**Fig. 3 | Shape recognition. a** Workflow of shape recognition in the human visual system. The schematic diagram is adapted with permission from refs. 47,48. LGN lateral geniculate nucleus, V1 primary visual cortex, V2 secondary visual cortex, V4 extrastriate cortex, IT inferotemporal cortex, NN neural network. The hardware structure of sparse neural networks with an orientation selector. **b** Test photo-current results (marked with the gray shading in the background) for the convolution of the 50 × 50 pixel regular hexadecagon light input with eight 5 × 5 orientation convolution kernels (OCKs). **c** Schematic of the hardware principle for shape recognition. The light-mask input is generated by illuminating the light mask consisting of a right-angled triangle and the frosted glass. The right-angled triangle input is divided into five regions for OCK convolution, which are processed through the 5 × 8 × 4 sparse neural network. Simulation (**d**) and experimental conductivity (**e**) weights of the double-layer sparse neural network with 30 epochs. **f** Simulation (red) and experiment (blue) results of the recognition rate with 30 epochs. **g** Comparison of this sparse neural network and the fully connected neural network in the recognition rate, loss, device usage, and programming energy for one operation. The error bars represent the standard deviation.

approximately 61.1% from about 2.3 nJ to 0.9 nJ per operation (Fig. 3g, Supplementary Note 1).

## Motion tracking

The M pathway of the human visual system implements motion tracking based on the Barlow–Levick model[46], as illustrated in Fig. 4a. Light successively stimulates photoreceptors at different sites and transmits signals to the back end of axons at different time scales following the length-dependent signal transmission delay of axons. The CSRF-based direction selector in the retina and visual cortex only produces signal superposition and activation to the motion stimulus that is parallel to the given movement direction. The higher visual cortex receives the pre-level information and controls the eye movement to track moving objects.

The visual pathway-replicated hardware for motion tracking (Fig. S16) is constructed using one WSe$_2$ SFG array and related peripheral circuits for direction selection and controlling eye position in

accordance with the above neural circuit. The SFG array mimics the retina and works in photovoltaic diode mode. Meanwhile, the delay module, bidirectional shift register, and switch transistor replicate the mechanisms of the axon's delay, cortex's control, and eye movement, respectively (Supplementary Note 4).

First, the direction selector is constructed by the SFG array, delay modules, and AND gate. To simulate the biological delay mechanism (Fig. 4b), four adjacent rows (marked 1–4) of the SFG array are cascaded to the related delay module $\Delta t_{1-4}$ ($\Delta t_1 > \Delta t_2 > \Delta t_3 > \Delta t_4$, for a given direction "up"), which mimics axons of different lengths. When light stimulation is present on a row of the array, a photocurrent pulse is generated and converted into a photovoltage by the TIA. The temporal photovoltages ($V_{ph}$) are delayed by the respective delay modules to obtain the corresponding temporal voltage signals ($V_{dl}$). Then they are fed into the four-input AND gate to get an output voltage ($V_{out}$), which determines the direction. The movement direction "up" of the light stimulation generates a 40 ms timing $V_{ph}$ signal at the channel

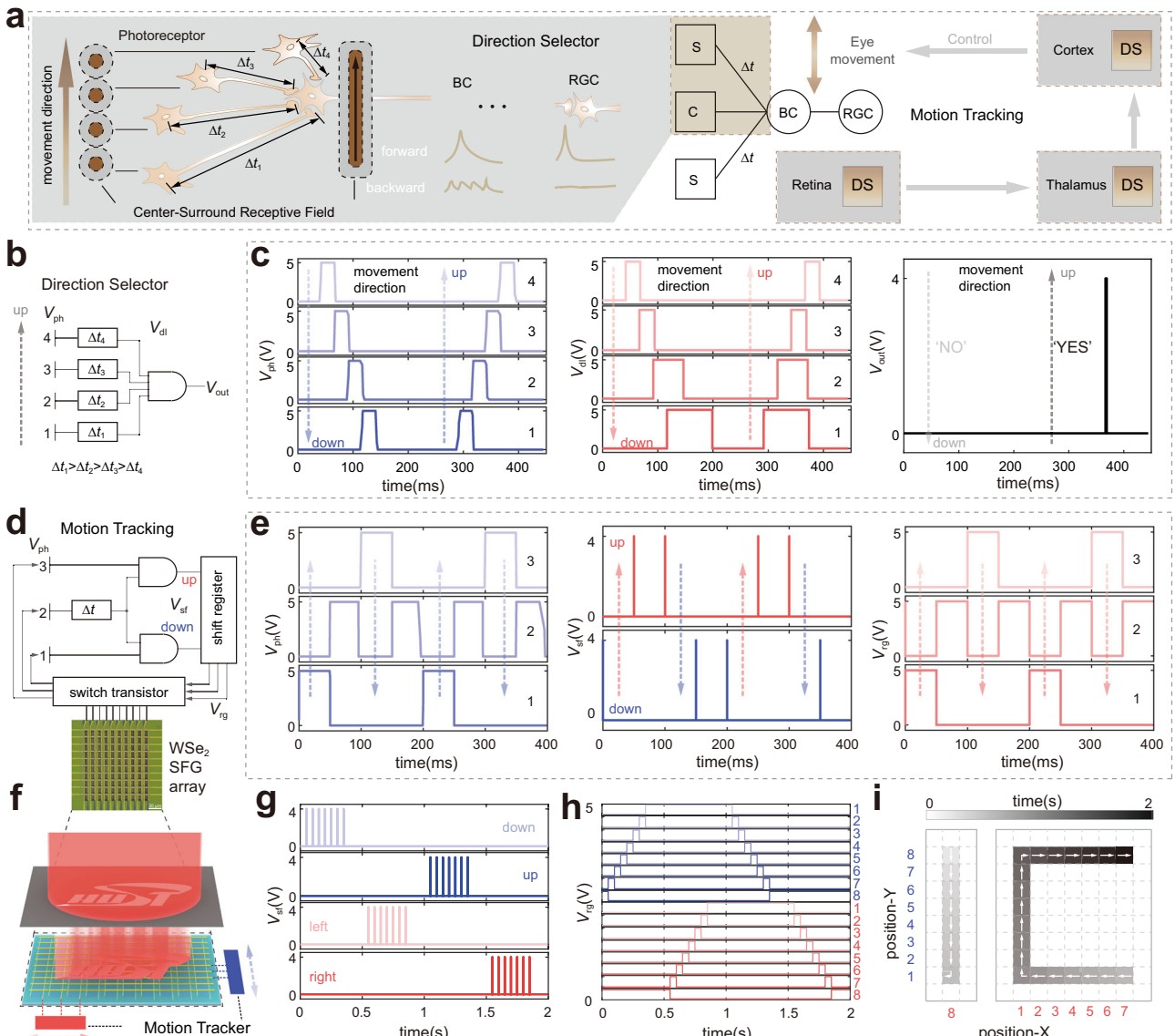

**Fig. 4 | Motion tracking. a** Principle diagram of the direction selector and motion tracking in the human visual system: center-surround receptive field (CSRF)-based direction selector with axonal connections of different lengths, and eye movement control by the cortex that processes previous visual information. The schematic diagram is adapted with permission from refs. 44,46. $\Delta t$: transmission delay of axons; BC bipolar cells, RGC retinal ganglion cells, S/C surround/center, DS direction selector. The gray shading on the left demonstrates the principle of center-surround (marked with the brown shading) direction selector based on the delay mechanism. The gray shadings on the right highlight DS in the retina, thalamus and cortex. **b** Workflow diagram of direction selector: photovoltaic voltage $V_{ph}$ of the front-end device array as input, converted into $V_{dl}$ by delay modules, and output voltage $V_{out}$ is summarized by a multiple-input AND gate. **c** Test results of $V_{ph}$, $V_{dl}$, and $V_{out}$ with light stimulus moving down and up at a speed of 20 pixels/s. **d** Workflow diagram of 1D bidirectional motion tracking: $V_{ph}$ as input, converted into up and down movement signals $V_{sf}$ through delay modules and AND gates, the $V_{rg}$ of the shift register is adjusted to control the switch transistors moving along the illumination site. **e** Test results of $V_{ph}$, $V_{sf}$, and $V_{rg}$ during the upward and downward movements of the light stimulus. **f** Schematic of motion tracking in a 2D plane with two 1D bidirectional motion tracker orthogonally arranged. The "HUST" mask moves clockwise along the "I"- and "C"-shaped light trajectory. Test results of $V_{sf}$ (**g**), $V_{rg}$ (**h**), and diagram of motion trajectory (**i**). Color bar, time for movement.

from 1 to 4. After the delay process, the movement in the given direction "up" generates high-level $V_{dl}$ signals in all four channels simultaneously to output a high-level $V_{out}$ that represents the right direction (Fig. 4c). Otherwise, the movement direction "down" cannot generate four high-level channel signals $V_{dl}$ at the same time, thus the output of the AND gate is always low, which is judged to be the wrong direction. Second, 1D bidirectional motion tracking is performed by adding a switch transistor array and a bidirectional shift register as the direction controller (Fig. 4d). The direction selector is a simplified version of a bidirectional selector, which receives $V_{ph}$ of the array in Channels 1–3 and outputs direction selective signal $V_{sf}$ from two dual-input AND gates to determine the movement direction "up" or "down"

for the shift register. Then the output signal $V_{rg}$ of the shift register automatically controls the gating of the switch transistor and accepts the $V_{ph}$ of the corresponding array channel according to the light stimulus's direction. When the light stimulus $V_{ph}$ shifts from 2 to 3 (1), the high-level signal $V_{sf}$ of the bidirectional selector is generated to determine the direction "up" ("down") and serves as the shift signal of the bidirectional shift register. The output of the shift register ($V_{rg}$) is controlled by $V_{sf}$, and the related switch transistor is selected along the light stimulation (Fig. 4e). Therefore, the function of motion tracking in the M pathway is replicated in the hardware. Finally, by adding the 1D bidirectional motion tracker in two orthogonal directions, motion tracking can be performed on a 2D plane (Fig. 4f). One is placed

horizontally to track "left" and "right" movement, and the other vertically to track "up" and "down" movement. The mark with the pattern of "HUST" is placed between the light source and the array sample (Fig. S1). When the mask moves clockwise along the trajectory of "I" and "C" at a speed of 20 pixels/s, the light stimulus induces the direction selector of "down," "left," "up," and "right" to produce output $V_{sf}$ sequentially (Fig. 4g). The shift register output sites (Fig. 4h) of these two motion tracker represent the position of the light stimulus, which coincide with the mask's moving trajectory (Fig. 4i). Therefore, the experiment setup demonstrates motion tracking function similar to the human visual pathway. As a prototype, the hardware can also support high-speed tracking tasks by precisely programming the time of delay.

## Discussion

In summary, we proposed visual pathway-replicated hardware to realize red–green color-blindness processing, low-power shape recognition, and self-driven motion tracking in the experiment, which are hardly achieved by the hardware that neglects the neural circuits of visual pathways. The unique $WSe_2$ SFG device in the crossbar array can function in photovoltaic diode and bipolar transistor modes with nonvolatile reconfigurable positive/negative optical responsivity and conductivity, making the array the core building block for retinal CSRF and the visual cortex's neural networks. This enhances the seamless integration of intricate human vision capabilities into a single chip, made possible by the innovation of the dual-mode functional device. With the feasibility of the heterogenous integration of 2D materials[49,50], this $WSe_2$ SFG array can be compatible with complementary metal oxide semiconductor-based peripheral circuits to build human visual pathway-replicated chips. In turn, it can inspire neuroscience and promote progress in the fields of driverless technology, brain–computer interfaces, and intelligent robots[51,52]. For example, the replication of the color visual pathway of human eyes can inherit color constancy[53], which has the potential to simplify the complexity of white balance in subsequent processing circuits and algorithms[54]. Furthermore, the complete reproduction of visual pathways will help develop brain-computer interface devices that are more adapted to neural structures, and help blind or color-blindness patients regain normal vision[55–57].

## Methods

### Material preparation
A ~$100 \times 100 \, \mu m^2$ $WSe_2$ flake was mechanically exfoliated from a bulk crystal (from HQ Graphene). The $WSe_2$ region in the $10 \times 10$ array was patterned by electron-beam lithography (EBL) and reactive ion etching (RIE) with $Ar/SF_6$ plasma.

### Device fabrication
First, we prepared a bottom gate (with an 800 nm gap) array on a silicon wafer (coated with 300-nm-thick $SiO_2$) substrate by EBL and electron-beam evaporation (EBE) with Cr/Au (5 nm/25 nm). Secondly, the blocking layer of a 30-nm-thick $Al_2O_3$ gate oxide was deposited by atomic layer deposition (ALD), followed by EBE of 5 nm Pt as the floating layer. Thirdly, the floating gate region was defined by EBL, and the Pt layer was etched using RIE with Ar. Next, the tunneling layer of an 8-nm-thick $Al_2O_3$ was deposited by ALD. Then, the $WSe_2$ array was transferred to the desired site of the substrate using a PDMS (polydimethylsiloxane)/PVA (Polyvinyl Alcohol) assisted fixed-point transfer method. Part of the source-drain Cr/Au (5 nm/70 nm) electrodes was prepared by EBL and EBE. A 50-nm-thick $Al_2O_3$ isolation layer was deposited by ALD, and via holes through the $Al_2O_3$ isolator and gate pads were defined by EBL and etched by inductively coupled plasma RIE (ICP-RIE) with $BCl_3/Ar$ plasma. The other part of the second layer of the source-drain Cr/Au (5 nm/ 70 nm) electrodes was prepared by EBL and EBE. Finally, the sample was placed on a 40-pin chip holder and wire-bonded.

### Experimental setup
The schematic diagram of the experimental setup is shown in Fig. S1. A continuous wavelength (CW) laser (Yangtze Soton Laser, SC–PRO) was modulated to a specific wavelength and intensity through the monochromator and attenuator driven by a controller. Then the laser irradiated on the device array through a 10× microscope objective and a mask controlled by a stepper motor, to construct dynamic light information of different colors, intensity, and shapes. In the shape recognition section, the mask consists of a mask and the frosted glass to generate disordered right-angle triangle light input. The gate voltage pulses were supplied row by row switched by a column Ground line using waveform generators (National Instruments, PXI–5404). The output current signals were measured by source meters (National Instruments, PXIe–4141), or amplified and converted to voltage signals by trans-impedance amplifiers (TIAs) on the Printed Circuit Board Assembly (PCBA). The output voltage signals were recorded with source meters (National Instruments, PXIe–4142) or oscilloscopes (Tektronix, TDS2024). The basic optoelectronic measurement for unit devices was conducted in a Lakeshore probe station using a semiconductor analyzer (Keysight B1500A), 455/520/637 nm laser diode, and temperature controller (Thorlabs, ITC4020). The entire test was carried out in an atmospheric environment.

## Data availability
All data needed to evaluate the findings of this study are available within the Article. Source Data file has been deposited in Figshare under accession code https://doi.org/10.6084/m9.figshare.26234561[58].

## Code availability
All codes used in this study are available from the corresponding author (L. Y.) upon request.

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

## Acknowledgements

This work was co-funded by the National Natural Science Foundation of China (Grant nos. 62222404, 62304084 and 92248304) (L. Y.), the National Key Research and Development Plan of China (Grant nos. 2021YFB3601200) (L. Y.), the Major Program of Hubei Province (Grant no. 2023BAA009) (L. Y.), and the Research Grants Council of Hong Kong Postdoctoral Fellowship Scheme (Grant no. PDFS2223-4S06) (L. T.).

## Author contributions

Z. P. and L. Y. conceived the project. Z. P. designed the hardware. Z. P., L. X., X. H. and Z. L. fabricated the hardware. Z. P., X. Y., X. M. and X. H.

performed the measurement. L. X. and H. H. completed the characterization of the material. W. S. polished the article. L. Y. and L. T. provided key suggestions to optimize data expression. Z. P. and L. Y. wrote the manuscript. G. Y., S. L., T. J. and X. M supervised the project. All the authors discussed the results and reviewed the manuscript.

## Competing interests

The authors declare no competing interests.
