## [Peer Review file · Nature Communications]

Multifunctional human visual pathway-replicated hardware based on 2D materials

Corresponding Author: Professor Lei Ye

This manuscript has been previously reviewed at another journal. This document only contains reviewer comments, rebuttal and decision letters for versions considered at Nature Communications.

Version 0:

Reviewer comments:

Reviewer #1

(Remarks to the Author)

In the manuscript, the authors propose a human visual pathway-replicated hardware architecture based on a crossbar array consisting of split-float-gating 2D WSe₂ devices. The device features reconfigurable and non-volatile positive/negative optical responsivity and electrical conductivity, allowing it to mimic functions of the retina and visual cortex, such as color processing, shape recognition, and motion tracking. The corresponding peripheral circuits based on the split-float-gating 2D WSe₂ arrays were designed to demonstrate these functions. Even though these demonstrations are intriguing, I suggest the article undergo major revisions before publication, as many explanations and descriptions lack sufficient detail, affecting the readability of the article. Some comments are as follows:

1. The responsivity of the 2D array follows a Gaussian distribution. The non-uniform distribution requires each individual device to be trained separately, which increases the complexity and power consumption of the operation. Can the author explain how the size of the standard deviation affects the image processing performance of the devices?
2. The dependency between responsivity and gate voltage is not linear, resulting in the use of only limited gate voltage values to regulate the kernel. How does this impact image processing? How can the design of the device address this non-linear dependency issue?
3. What are the values of the convolution kernels used in each convolution operation? For example, what are the kernel values for the operations "R+, R-, G+, G-, B+, B-"?
4. In Fig. 2b, the kernel sizes of small RF and large RF are 3 × 3 and 5 × 5, respectively. But it seems the effective kernels in the two RF are all 3 × 3. It should be explained more detailly. In Fig. 2c, why is the VSD for green negative in G+R-, while the VSD for red is positive?
5. The descriptions of each figure in the text are not detailed enough, making it difficult for readers to understand what the figures are illustrating. For instance, what does the light-mask input in Fig. 3c refer to? Has a right triangle been processed in some way?
6. What do the labels [T1 T2 T3 T4]T refer to? Does "The right triangle pattern is divided into five regions" mean that a single right triangle is divided into five regions, or that the dataset of 1600 right triangles is divided into five regions? Is the division random or systematic?
7. How were the experimental results shown in Fig. 3e measured? How was the programming energy of 0.9 nJ calculated? The calculation methods for the various performance parameters in Fig. 3g need to be provided.
8. A more detailed description of the motion tracking process is needed. Especially for each figure, the description needs to be more detailed, rather than just simply describing some results in the text.

Reviewer #2

(Remarks to the Author)

This manuscript addresses artificial photoreceptive devices inspired by the visual pathway of the human eye. The authors developed WSe₂-based split floating gates (SFG) in the form of crossbar arrays, which mimic the functions of the retina and visual cortex. Additionally, with some peripheral circuits, the authors successfully demonstrated color blindness processing, shape recognition, and self-driven motion tracking. This paper is considered to be of high quality. However, a few areas require more detailed explanation.

Firstly, the complexity of the visual pathway in the human eye, as implemented in this manuscript, appears to be the most critical aspect. However, performing complex functions at the system level does not necessarily guarantee the novelty of the research. Therefore, it is necessary to discuss whether the WSe₂-based SFG is essential for performing these complex functions. Recent studies have integrated memristors with silicon or GaAs-based photodiode arrays to achieve imaging and neuromorphic characteristics, and there are many studies using various other materials as well.

Despite many interesting demonstrations, there does not appear to be any video demonstration to verify these results. If there are no particular difficulties, including a demo video for more intuitive understanding would significantly enhance the value of the paper.

The color blindness processing experiment seems to be an original experiment not encountered in previous research. However, the importance or significance of this study is not discussed in detail in the paper, making it difficult to understand its depth. Is there a specific direction in which this type of processing can be utilized in robotic vision? Comments on this aspect are necessary.

Reviewer #3

(Remarks to the Author)

In this work, Ye et al. designed and fabricated split-floating gate 2D tungsten diselenide crossbar array and human visual pathway-replicated hardware with rich functions such as color processing, shape recognition and motion tracking. This work is the first to show the cause of red–green color blindness at the circuit level, which is expected to inspire the common development of information devices and biological neuroscience. In shape recognition, the sparsity of visual pathway is replicated by extracting local features through convolution and introducing them into sparse neural networks, which effectively reduces power consumption and device usage. The self-driven motion tracking is realized by introducing Barlow-Levick model into the peripheral circuits. This work has novel and ingenious ideas, and the hardware has highly replicated the structure and function of the visual pathways, which opens up a new application prospect for the development of 2D materials in the field of image sensors. The manuscript is well written and in high quality. I believe it meets the high impact of Nature Communications and recommend publication after a minor revision. Here are some comments.

1. The authors prepared 2D WSe₂ transistor with the ON/OFF ratio of 107 (Fig.S2b), showing excellent semiconductor performance. In order to ensure the reliability of the material, could you provide Raman spectrum test results to verify its type and quality?
2. The authors used 2D WSe₂ of symmetrical bipolar type. As far as I know, the conductive type of WSe₂ varies with thickness. Please show me relevant thickness data of WSe₂.
3. In the section “Color processing”, the authors have creatively expressed the visual processing scenes of red–green color blindness through circuits. Could you explain how the color visual defect was imported in the experiment specifically?
4. In the section “Shape recognition”, the hardware can perform convolution to extract local features and import them in a double-layer sparse neural network for image classification. What’s authors’ consideration about the choice of sparsity and activation function?
5. In the section “Motion tracking”, the authors choose the Barlow–Levick model as the basic mechanics of human dynamic vision. I do think it’s suitable in this application for self-driven motion tracking, please specifically explain its principles and advantages.

Reviewer #4

(Remarks to the Author)

Version 1:

Reviewer comments:

Reviewer #1

(Remarks to the Author)

The authors addressed most of my concerns during the first round of review, but some descriptions in the manuscript are still unclear, which may affect the readability of the manuscript.

In the color processing section, Fig. 2a shows the distributions of red, green, and blue colors at different spatial locations. I have a few questions: Is the convolution of these three colors with the responsivity convolution kernel performed independently or simultaneously? Additionally, does the responsivity convolution kernel change depending on the color being processed? There is also an inconsistency between the convolution kernel values mentioned in the main text and the supplementary information, which needs clarification.

Regarding the shape recognition demonstration, the reviewer remains unclear about how the demonstration was conducted. Specifically, for “The light-mask input is generated by illuminating the light mask of a right-angle triangle.”, why does

illuminating a right-angle triangle mask generate a disordered light-mask input? Was this light-mask input randomly selected from a database of 1,600 samples? What was the basis for selecting the five OCKs shown in Fig. 3c, including two at 0°, two at 90°, and one at 45°? Is the discrete neural network processing applied only to the five parts within the red dashed box of the light-mask input? If so, why was a 15 × 15 pixels size light-mask input necessary? What is the purpose of the other light mask patterns shown in Fig. S1?

I request the authors to provide more detailed explanations for each demonstration, as this would help readers better understand the application potentials of this retina-mimicked hardware.

Reviewer #2

(Remarks to the Author)

The revised version is almost ready for publication.

Reviewer #3

(Remarks to the Author)

The authors have well addressed my comments in the revised manuscript, i have no further question on this manuscript and i am happy to recommend this great work for publication in Nature Communications.

Reviewer #4

(Remarks to the Author)

Response to REVIEWER COMMENTS

Response to Reviewer #1

General comments:

“In the manuscript, the authors propose a human visual pathway-replicated hardware architecture based on a crossbar array consisting of split-float-gating 2D WSe₂ devices. The device features reconfigurable and non-volatile positive/negative optical responsivity and electrical conductivity, allowing it to mimic functions of the retina and visual cortex, such as color processing, shape recognition, and motion tracking. The corresponding peripheral circuits based on the split-float-gating 2D WSe₂ arrays were designed to demonstrate these functions. Even though these demonstrations are intriguing, I suggest the article undergo major revisions before publication, as many explanations and descriptions lack sufficient detail, affecting the readability of the article.”

Response:

Thank you very much for your recognition and support. We have responded carefully to your valuable questions and have made more specific treatment of the details of the content of the manuscript. Your questions have been very important to the optimization of the manuscript, and we hope the answers meet your expectations.

Comment 1:

“The responsivity of the 2D array follows a Gaussian distribution. The non-uniform distribution requires each individual device to be trained separately, which increases the complexity and power consumption of the operation. Can the author explain how the size of the standard deviation affects the image processing performance of the devices?”

Response:

Thank you very much for your comments! In our human visual pathway-replicated hardware, each individual device is inevitable to be trained separately to replicate the center-periphery antagonism mechanism of human eyes, as the response of Gaussian

distribution is a conventional method^{1,2}. In the test experiments, the used convolution kernels are only small sizes of 3×3 and 5×5 , the weight values required are limited and fixed. Only a small number of discrete values need to be selected according to the Gaussian function, so the complexity is limited. The device is also modulated by the non-volatile floating gate, which can reduce the energy consumption of programming.

The Gaussian standard deviation σ determines the degree of blur of the Gaussian function to the image processing, in other words, regulates the effect of the surrounding on the central signal suppression in the center-periphery antagonism. The larger the σ is, the fuzzier the image processing information is, which is represented as extracting global information and constructing a large CSRF. The smaller the σ , it mainly carries the information of each independent pixel, showing more details and constructing a small CSRF.

The corresponding explanation in the revised manuscript is located at line 160-165. The revised parts are also listed below:

“The Gaussian standard deviation σ determines the degree of blur in image processing, a larger one is fuzzier to extract global information and a small one is more concentrated to get local information. The $\sigma = 0.08$ and $\sigma = 0.12$ respectively set the responsivity distribution of small (3×3) and large (5×5) CSRF with two values (0.04 and -0.008 A/W) and three values (0.007, 0.003 and -0.0001 A/W) (Figs. 2b and S13).”

References:

1. Wang, C. Y. et al. Gate-tunable van der Waals heterostructure for reconfigurable neural network vision sensor. *Sci. Adv.* **6**, aba6173 (2020).
2. Zhang, Z. et al. All-in-one two-dimensional retinomorph hardware device for motion detection and recognition. *Nat. Nanotechnol.* **17**, 27–32 (2022).

Comment 2:

“The dependency between responsivity and gate voltage is not linear, resulting in the use of only limited gate voltage values to regulate the kernel. How does this impact image processing? How can the design of the device address this non-linear

dependency issue?”

Response:

We are grateful for your valuable comments. In this work, the non-linear dependency does not impact the imaging processing, as addressed from two aspects. Firstly, the non-linear dependency between responsivity and gate voltage does impact the accuracy of image processing only in the situation where a lot of different weights is required. But our device only needs to be set to a small number of fixed discrete values (two for small CSRF and three for large CSRF) to construct the optical responsivity matrix, therefore the quasi-linear dependency is sufficient to meet the experimental setup requirements. This is also demonstrated in other recent work¹. Secondly, only the part between -0.04 to 0.04 A/W is used, which is nearly linear and allows for tuning the limited convolution kernels to suppress the impact of nonlinearity².

As for methods to address the non-linear dependency issue, there are some improvement methods in device design. Generally, the electrode contact (between the source/ drain and the material), the interface quality (between the channel material and the gate dielectric) are often considered as the primary factors affecting the linearity of the device. Therefore, optimization of device design can be focused on improving the electrode contact and interface quality. The electrode contact can be improved by using suitable contact metal³ or transfer electrodes⁴. The interface quality can be enhanced by an improved dielectric preparation process⁵.

The corresponding explanation in the revised manuscript is located at line 119-121. The revised parts are also listed below: “Here, the impact of non-linear dependency on subsequent imaging processing is suppressed by fixing the weights within several discrete values, also as demonstrated in the recent work²⁵.”

References:

1. Mennel, L. et al. Ultrafast machine vision with 2D material neural network image sensors. *Nature* **579**, 62–66 (2020).
2. Pi, L. et al. Broadband convolutional processing using band-alignment-tunable heterostructures. *Nat. Electron.* **5**, 248–254 (2022).
3. Shen, PC., Su, C., Lin, Y. et al. Ultralow contact resistance between semimetal and monolayer

semiconductors. *Nature* **593**, 211–217 (2021).

4. Liu, Y., Guo, J., Zhu, E. et al. Approaching the Schottky–Mott limit in van der Waals metal–semiconductor junctions. *Nature* **557**, 696–700 (2018).

5. Zeng, D., Zhang, Z., Xue, Z. et al. Single-crystalline metal-oxide dielectrics for top-gate 2D transistors. *Nature* (2024). <https://doi.org/10.1038/s41586-024-07786-2>

Comment 3:

“What are the values of the convolution kernels used in each convolution operation? For example, what are the kernel values for the operations ‘R+, R-, G+, G-, B+, B-’?”

Response:

Thank you very much for your comments! For color processing (Fig. 2b, S13), the small (3×3) kernel has two values (0.04 and 0.008 A/W) and the large (5×5) CSRF has three values (0.007, 0.003 and 0.0001 A/W), constructing convolution kernels as below:

$$\begin{bmatrix} 0.008 & 0.008 & 0.008 \\ 0.008 & 0.04 & 0.008 \\ 0.008 & 0.008 & 0.008 \end{bmatrix}$$

$$\begin{bmatrix} 0.0001 & 0.0001 & 0.0001 & 0.0001 & 0.0001 \\ 0.0001 & 0.003 & 0.003 & 0.003 & 0.0001 \\ 0.0001 & 0.003 & 0.007 & 0.003 & 0.0001 \\ 0.0001 & 0.003 & 0.003 & 0.003 & 0.0001 \\ 0.0001 & 0.0001 & 0.0001 & 0.0001 & 0.0001 \end{bmatrix}$$

In each site, the responsivity value for “R+, R-, G+, G-, B+, B-” operations in the 3×3 or 5×5 CSRF is the same above, where the sign “+” / “-” represents potentiated/depresed information at the corresponding site.

For shape recognition (Fig. 3b), the convolution kernel values (for example, OCK at 0°) are shown as below:

$$OCK_{0^\circ} = \begin{bmatrix} -0.02 & -0.02 & -0.02 & -0.02 & -0.02 \\ -0.02 & -0.02 & -0.02 & -0.02 & -0.02 \\ +0.1 & +0.1 & +0.1 & +0.1 & +0.1 \\ -0.02 & -0.02 & -0.02 & -0.02 & -0.02 \\ -0.02 & -0.02 & -0.02 & -0.02 & -0.02 \end{bmatrix}$$

The OCKs in eight directions are constructed by giving “+” 0.1 A/W and “-” -0.02 A/W, as shown in Fig. 3b.

The corresponding explanation is added in the supplementary information (Supplementary Note 2).

Comment 4:

“In Fig. 2b, the kernel sizes of small RF and large RF are 3×3 and 5×5 , respectively. But it seems the effective kernels in the two RF are all 3×3 . It should be explained more detailly. In Fig. 2c, why is the V_{SD} for green negative in $G+R-$, while the V_{SD} for red is positive?”

Response:

Thanks very much for your comments.

The large RF in Fig.2b is indeed 5×5 , but it cannot be distinguished in color due to the magnitude difference in the outermost responsiveness. We have replaced them with plots that mark underlying data in the revised manuscript and showed the original data in Source Data.

The V_{SO} for green is positive and for red is negative in $G+R-$, we have corrected the figure to avoid misunderstanding in the revised manuscript.

The corresponding figure is changed in the revised manuscript (Fig.2).

Comment 5:

“The descriptions of each figure in the text are not detailed enough, making it difficult for readers to understand what the figures are illustrating. For instance, what does the light-mask input in Fig. 3c refer to? Has a right triangle been processed in some way?”

Response:

Sincere thanks for your comments. We have added more detailed descriptions for all figures in the revised manuscript.

The light-mask input in Fig. 3c refers to the transmitted light pattern when the light illuminates the hollow mask of right-angled triangle (Fig. S1). The transmitted light pattern is quantified to the 15×15 -pixel scale as input. In the experiment, the input is first divided, and five 5×5 -pixel regions of spatial distribution are chosen as the

illumination input by moving the mask, and respectively convolved with five specific OCKs.

Some expansions in the text and figure notes have been made in the revised manuscript, as listed below:

“Specifically, the conductance value of the WSe₂ transistor connecting the R–G SO and DO is set to 100 nS, while the other channels are set to 1 μS.” (line 182–183)

“The light illuminates the light mask of a right-angled triangle (Fig. S1), and the transmitted light pattern is quantified to the 15 × 15–pixel size as the light-mask input.” (line 209–211)

“In the experiment, the light input state of these five regions is adjusted by moving the position of the mask (Fig. S1).” (line 213–215)

“The light-mask input is generated by illuminating the light mask of right-angled triangle.” (line 520–521)

Comment 6:

“What do the labels [T₁ T₂ T₃ T₄] T refer to? Does ‘The right triangle pattern is divided into five regions’ mean that a single right triangle is divided into five regions, or that the dataset of 1600 right triangles is divided into five regions? Is the division random or systematic?”

Response:

Thanks very much for your comments. The “T” in labels [T₁ T₂ T₃ T₄] refers to the tag of sample in the database, and the tag of right triangles with four different orientations (Fig. S1) are encoded into this 1 × 4 vector by one-hot encoding. Here, each pattern of the right triangle is spatially divided into five regions by pixel (Supplementary Note 3). The division of five regions per sample is systematic according to corresponding position.

Comment 7:

“How were the experimental results shown in Fig. 3e measured? How was the programming energy of 0.9 nJ calculated? The calculation methods for the various

performance parameters in Fig. 3g need to be provided.”

Response:

We thank you for your comments.

Fig. 3e shows the experimental conductivity weights of the double-layer sparse neural network with 30 epochs, which are mapped by the trained weight values in simulation at first. Then the actual conductivity value is set according to the number of V_G pulses and is measured through output curves after setting.

The evaluation method and calculation formula of programming energy are given in 7) of Supplementary Note 1. “During the programming process, the drain and source are grounded. The energy consumption only originates from the voltage pulse supplied to the gate. The voltage pulse is fixed at 1 V and 10 ms, with a gate leakage current of less than 0.1 nA. Therefore, the programming energy is estimated to be no more than 1 pJ/spike, calculated by $E_{\text{program}} = V_{\text{pulse}} \cdot I_{\text{pulse}} \cdot t_{\text{pulse}}$. The programming energy of the sparse neural network and the fully connected neural network (Fig. 3g) is estimated by performing 32 spikes on average per device for 64-level weights.” Therefore, the programming energy per device is evaluated to 32 pJ. It consumes about nearly 0.9 nJ with 28 devices for sparse NN, while 2.3 nJ with 72 devices for full-NN.

The recognition rate and loss in Fig. 3g are the statistical results of simulation (Fig. S15). The calculation method of device usage and programming energy is added to 7-8) of Supplementary Note 1. “Device usage depends on the numbers of weights in the NN. For this $5 \times 8 \times 4$ NN, full connected one has $5 \times 8 + 8 \times 4 = 72$ weights, sparse connected one has only 28 weights (Fig. 3c, Source Data 3e).”

Comment 8:

“A more detailed description of the motion tracking process is needed. Especially for each figure, the description needs to be more detailed, rather than just simply describing some results in the text.”

Response:

Thank you very much for your advice. We have detailed the relevant description in the revised manuscript, which is marked blue. The revised parts are listed below:

“Light successively stimulates photoreceptors at different sites and transmits signals to the back end of axons at different time scales following the length-dependent signal transmission delay of axons.” (line 227–229)

“First, the direction selector is constructed by the SFG array, delay modules, and AND gate.” (line 241–242)

“which mimics axons of different lengths. When light stimulation is present on a row of the array, a photocurrent pulse is generated and converted into a photovoltage by the TIA. The temporal photovoltages (V_{ph}) are delayed by the respective delay modules to obtain the corresponding temporal voltage signals (V_{dl}). Then they are fed into the four-input AND gate to get an output voltage (V_{out}), which determines the direction. The movement direction “up” of the light stimulation generates a 40 ms timing V_{ph} signal at the channel from 1 to 4.” (line 244–250)

“Otherwise, the movement direction “down” cannot generate four high-level channel signals V_{dl} at the same time, thus the output of the AND gate is always low, which is judged to be the wrong direction.” (line 253–255)

“The direction selector is a simplified version of a bidirectional selector, which receives V_{ph} of the array in Channels 1–3 and outputs direction selective signal V_{sf} from two dual-input AND gates to determine the movement direction “up” or “down” for the shift register. Then the output signal V_{rg} of the shift register automatically controls the gating of the switch transistor and accepts the V_{ph} of the corresponding array channel according to the light stimulus’s direction.” (line 257–262)

“Therefore, the function of motion tracking in the M pathway is replicated in the hardware.” (line 267–268)

“One is placed horizontally to track “left” and “right” movement, and the other vertically to track “up” and “down” movement. The mark with the pattern of “HUST” is placed between the light source and the array sample (Fig. S1).” (line 270–272)

“The shift register output sites (Fig. 4h) of these two motion tracker represent the position of the light stimulus, which coincide with the mask’s moving trajectory (Fig. 4i).” (line 275–277)

Response to Reviewer #2

General comments:

“This manuscript addresses artificial photoreceptive devices inspired by the visual pathway of the human eye. The authors developed WSe₂-based split floating gates (SFG) in the form of crossbar arrays, which mimic the functions of the retina and visual cortex. Additionally, with some peripheral circuits, the authors successfully demonstrated color blindness processing, shape recognition, and self-driven motion tracking. This paper is considered to be of high quality. However, a few areas require more detailed explanation.”

Response:

Thank you very much for your recognition of the quality of this work. We provide detailed peer-to-peer responses to your professional questions after thoughtful, extensive research and comparative analysis. I hope the responses meet your expectations.

Comment 1:

“Firstly, the complexity of the visual pathway in the human eye, as implemented in this manuscript, appears to be the most critical aspect. However, performing complex functions at the system level does not necessarily guarantee the novelty of the research. Therefore, it is necessary to discuss whether the WSe₂-based SFG is essential for performing these complex functions. Recent studies have integrated memristors with silicon or GaAs-based photodiode arrays to achieve imaging and neuromorphic characteristics, and there are many studies using various other materials as well.”

Response:

Thank you very much for your comments. Here, we discuss the necessity and innovation for WSe₂ SFG device and array to implement the complex functions of the visual pathway.

Firstly, complex functions of the visual pathway require the core building block for retinal CSRF and the visual cortex's neural networks. The WSe₂ SFG array can realize both functions in the same device structure (memtransistor + diode transistor +

photodetector all-in-one), which is more compatible with visual pathway. But other schemes, like integrating memristors with photodetectors¹⁻³, separate these functions into different device components. Besides, the bipolarity of WSe₂ is also necessary to realize self-powered photovoltaic effect and simulate antagonistic mechanisms in the human visual system. This is complex to achieve for unipolar materials, and chemical doping or building heterostructures also sacrifice the reconfigurability. The simplified process complexity promising the reduction cost of monolithic integration, and functional reconfigurability is more competent than other heterogeneous material systems.

Secondly, recent works (summarized in Table R1) have presented some human-like functions, including the retina¹⁻⁴ and the retina/ visual cortex⁵⁻⁶. The relatively limited functions are demonstrated because they have only considered the device design to mimic retina and the abstract network structure of visual cortex. Inspired by the actual connection structures and functions of visual pathways from the retina to the visual cortex, our work has fabricated the visual pathway-replicated hardware based on the cross-bar array consisting of the unit device with two modes to build more real and abundant functions of human visual system. Specifically, some unique functions of human visual system are implemented in our hardware: i) real color vision based on antagonistic mechanisms; ii) shape recognition based on spatial CSRF and sparse connected neural networks; iii) self-driven motion tracking based on delayed superposition of axon signals.

The corresponding explanation related to this comment has been added to the revised manuscript, located at discussion section (line 288-289). The revised part is also listed below: “This enhances the seamless integration of intricate human vision capabilities into a single chip, made possible by the innovation of the dual-mode functional device.”

Table R1: Visual hardware comparison

	Ref. 1	Ref. 2	Ref. 3	Ref. 4	Ref. 5	Ref. 6	This work
Biological System	Retina	Retina	Retina	Retina	Retina +	Retina +	Retina, visual cortex, visual

emulation					visual cortex	visual cortex	pathways
Unit Device structure	Si PD + RRAM	1P1R (InGaAs PD, HfO _x ReRAM)	GaAs PD + Si RRAM	NbS ₂ /MoS ₂ phototransistor	(O/E) RRAM	MoS ₂ /h-BN/WSe ₂ phototransistor	WSe ₂ SFG photodiode /transistor
Working mode	Seperated	Seperated	Seperated	In-sensor computing	In-sensor computing	In-sensor computing	In-sensor computing
Color vision	N/A	N/A	N/A	N/A	N/A	Tricolor	Color blindness
Shape vision	N/A	Image classification	Image classification	Image recognition	Image recognition	Image recognition	Shape recongnition
Dynamic vision	Spatial smoothing	N/A	N/A	Trace extraction	N/A	N/A	Self-driven motion tracking
Recognition rate	N/A	82%	N/A	92%	97.3%	100%	> 96%
Power consumption	7.8 μ W /pathway	38.51 μ W /cycle	2.5 mW /chip	0.42 pJ/pulse	151.579 TOPS/W	2.4 \times 10 ⁻¹⁷ J/spike	< 0.77 nJ/op

References:

1. Eshraghian, J. K. *et al.* Neuromorphic Vision Hybrid RRAM-CMOS Architecture. *IEEE Trans. Very Large Scale Integr. Syst.* **26**, 2816–2829 (2018).
2. Lee, D. *et al.* In-sensor image memorization and encoding via optical neurons for bio-stimulus domain reduction toward visual cognitive processing. *Nat. Commun.* **13**, 1–9 (2022).
3. Choi, C. *et al.* Reconfigurable heterogeneous integration using stackable chips with embedded artificial intelligence. *Nat. Electron.* **5**, 386–393 (2022).
4. Huang, P. Y. *et al.* Neuro-inspired optical sensor array for high-accuracy static image recognition and dynamic trace extraction. *Nat. Commun.* **14**, 1–9 (2023).
5. Zhou, G. *et al.* Full hardware implementation of neuromorphic visual system based on multimodal optoelectronic resistive memory arrays for versatile image processing. *Nat. Commun.* **14**, 1–11 (2023).
6. Zhang, T. *et al.* High performance artificial visual perception and recognition with a plasmon-enhanced 2D material neural network. *Nat. Commun.* **15**, 1–10 (2024).

Comment 2:

“Despite many interesting demonstrations, there does not appear to be any video demonstration to verify these results. If there are no particular difficulties, including a

demo video for more intuitive understanding would significantly enhance the value of the paper.”

Response:

Thanks a lot for your valuable advice. We have prepared a demo video attached to the supplementary files (Supplementary Movie). Since there are many test contents in this work, and most of them are the same repeated operations, this video shows the typical mask photoelectric scanning test.

The demo video consists of two parts. The first part introduces the composition of optoelectronic measurement system and the whole process of experimental setup (also in Fig. S1). The composition and connection status of five main modules including continuous wavelength (CW) laser, monochromator/attenuator, stepper motor-controlled mask, sample, NI PXIe are shown. The second part is a recorded measurement video. First, the state of the sample and the laser light source is pre-regulated. Then, NI is started, the position of the mask is controlled by the stepper motor controller, and the sample current signal is collected.

Comment 3:

“The color blindness processing experiment seems to be an original experiment not encountered in previous research. However, the importance or significance of this study is not discussed in detail in the paper, making it difficult to understand its depth. Is there a specific direction in which this type of processing can be utilized in robotic vision? Comments on this aspect are necessary.”

Response:

Thank you very much for your advice. The color processing experiment seems to have two development prospects in computer vision/robot vision.

Firstly, color constancy is one of the amazing abilities of perceptual constancy of the human visual system (HVS), which enables the perceived color of objects largely constant as the light source color changes. In contrast, captured with regular digital cameras or videos, the physical color of scenes may be shifted by the varying external illuminant. One of the fundamental requirements in computer vision, especially for the

robust color-based systems (e.g., color-based object recognition and tracking), is to extract reliable color cues that are invariant to the changes in external lighting¹. A common solution is to first estimate the scene illuminant, which is then used to correct the color-biased images to get the so-called canonical images and also known as “white balance”. Traditional cameras only receive the intensity information of the three components of RGB to obtain the color information, and additional compensation algorithms are required for color constancy, which depend only on the statistical distributions of individual pixels and ignore their spatial contexts. However, replicating visual pathway of color vision in HVS, the hardware in our work could capture the strong dependencies between nearby pixels using the concentrically organized center-surround structure with both spectral and spatial opponency of the DO cells². Therefore, the function of color constancy could be arranged in the hardware at the preprocessing period, which will simplify the complexity of subsequent processing circuits and algorithms.

Secondly, for color-blindness patients, due to the weakening or obstruction of some color vision in related visual pathway, color cannot be accurately recognized, which will seriously affect the daily life. By understanding the causes of color blindness in the visual processing pathway at the hardware level, we can reverse design stimulus-related neural signals, and hopefully eliminate color blindness through brain-computer interface³⁻⁴. For the blind, artificial eyes that highly simulate visual pathways can reduce the additional circuit consumption of functions such as white balance, and can better integrate with the human visual system. Following this thought and designing hardware that is compatible with the human visual system, damaged neural structures can be replaced to stimulate the vision⁵, which is similar to Elon Reeve Musk and Neuralink's next generation product “Blindsight”. It can be used to build electronic prosthetic eyes that match the visual system and help blind or color-blind patients regain normal color vision.

The corresponding discussions related to this comment have been added in the revised manuscript, located at discussion section. The revised parts are also listed below: “For example, the replication of the color visual pathway of human eyes can inherit

color constancy⁵⁰, which has the potential to simplify the complexity of white balance in subsequent processing circuits and algorithms⁵¹. Furthermore, the complete reproduction of visual pathways will help develop brain-computer interface devices that are more adapted to neural structures, and help blind or color-blindness patients regain normal vision⁵²⁻⁵⁴.” (line 294–299)

References:

1. Gijsenij, A., Gevers, T. & Van De Weijer, J. Computational color constancy: Survey and experiments. *IEEE Trans. Image Process.* **20**, 2475–2489 (2011).
2. Gao, S. B., Yang, K. F., Li, C. Y. & Li, Y. J. Color Constancy Using Double-Opponency. *IEEE Trans. Pattern Anal. Mach. Intell.* **37**, 1973–1985 (2015).
3. Luo, X. et al. A bionic self-driven retinomorphoc eye with ionogel photosynaptic retina. *Nat. Commun.* **15**, 1–9 (2024).
4. Yang, R. et al. Assessment of visual function in blind mice and monkeys with subretinally implanted nanowire arrays as artificial photoreceptors. *Nat. Biomed. Eng.* 1–22 (2023) doi:10.1038/s41551-023-01137-8.
5. Beauchamp, M. S. et al. Dynamic Stimulation of Visual Cortex Produces Form Vision in Sighted and Blind Humans. *Cell* **181**, 774–783 (2020).

Response to Reviewer #3

General comments:

“In this work, Ye et al. designed and fabricated split-floating gate 2D tungsten diselenide crossbar array and human visual pathway-replicated hardware with rich functions such as color processing, shape recognition and motion tracking. This work is the first to show the cause of red–green color blindness at the circuit level, which is expected to inspire the common development of information devices and biological neuroscience. In shape recognition, the sparsity of visual pathway is replicated by extracting local features through convolution and introducing them into sparse neural networks, which effectively reduces power consumption and device usage. The self-driven motion tracking is realized by introducing Barlow-Levick model into the peripheral circuits. This work has novel and ingenious ideas, and the hardware has highly replicated the structure and function of the visual pathways, which opens up a new application prospect for the development of 2D materials in the field of image sensors. The manuscript is well written and in high quality. I believe it meets the high impact of Nature Communications and recommend publication after a minor revision. Here are some comments.”

Response:

Thank you very much for your recognition and affirmation. In response to your questions, we have carefully considered and made detailed replies point by point. Hope to meet your expectation.

Comment 1:

“The authors prepared 2D WSe₂ transistor with the ON/OFF ratio of 10⁷ (Fig.S2b), showing excellent semiconductor performance. In order to ensure the reliability of the material, could you provide Raman spectrum test results to verify its type and quality?”

Response:

Thanks for your comments. Excited by 532-nm laser, the Raman spectra of WSe₂ in the array (Fig. R1) has two first-order Raman peaks ($E_{2g}^1 \sim 248.66$ and $A_{1g} \sim$

256.72) and three second-order modes (308.57, 372.50, 394.65), which are typical Raman peaks of WSe₂ and confirm its high quality^{1,2}.

Fig. R1 | Raman spectrum of WSe₂ in the array.

References:

1. Qian, Q. et al. Layer-dependent second-order Raman intensity of MoS₂ and WSe₂: Influence of intervalley scattering. *Phys. Rev. B* **97**, 165409 (2018).
2. Luo, X. et al. Effects of lower symmetry and dimensionality on Raman spectra in two-dimensional WSe₂. *Phys. Rev. B* **88**, 195313 (2013).

Comment 2:

“The authors used 2D WSe₂ of symmetrical bipolar type. As far as I know, the conductive type of WSe₂ varies with thickness. Please show me relevant thickness data of WSe₂.”

Response:

Thanks a lot for your suggestion. From the AFM data of WSe₂ material on the array (Fig.R2), the thickness is ~ 6.21 nm.

Fig. R2 | AFM amplitude mapping of WSe₂ in the array. The thickness of WSe₂ is 6.21 nm, as marked as a white dotted box. Scale bar, 2μm.

Comment 3:

“In the section “Color processing”, the authors have creatively expressed the visual processing scenes of red–green color blindness through circuits. Could you explain how the color visual defect was imported in the experiment specifically?”

Response:

Thank you very much for your comment. At the circuit level, we adjust the proportion of red-green antagonistic channels to 10%. Specifically, the conductance value of the WSe₂ transistor connecting the red-green antagonistic SO and DO is set to 100 nS, while the other channels are set to 1 μS.

The corresponding explanation in the revised manuscript is located at line 179-180.

Comment 4:

“In the section “Shape recognition”, the hardware can perform convolution to extract local features and import them in a double-layer sparse neural network for image classification. What’s authors’ consideration about the choice of sparsity and activation function?”

Response:

Thanks for your comment. In the section “Shape recognition”, we make full reference to the processing process of image information in human visual pathways. The choice of sparsity is optimized to match the process of combining local information in visual pathways. For the activation function, we choose a Mish function similar to the rectification characteristics of the device. This activation can be configured with the current device, and can result to a relatively higher recognition rate obtained by screening during simulation (Figs. S15d-e).

Comment 5:

“In the section “Motion tracking”, the authors choose the Barlow–Levick model as the basic mechanics of human dynamic vision. I do think it’s suitable in this application for self-driven motion tracking, please specifically explain its principles

and advantages.”

Response:

Thanks a lot for your valuable suggestion. The principles are discussed in detail. Barlow-levick model is a signal delay-based model^{1,2}, which abstracts the structure and function of mammalian motion sensing neural pathways. It consists of two input lines carrying the brightness signals which are compared after one of the signals has been delayed. The comparison is accomplished by a specific logical operation, an AND-NOT or veto gate, suppressing the detector's activity when the delay line is activated first and, consequently, both signals arrive simultaneously at the AND-NOT gate. For motion in the detector's preferred direction the veto signal arrives too late to have an effect¹. In this work, the authors replace the AND-NOT gate with the AND gate, so the delay from long to short is the preferred direction. According to the length of the delay time, the signal superposition state of different stimulus directions is determined, so as to judge the dynamic information of the motion.

The advantage lies in three aspects. Firstly, the model is highly compatible with the processing mechanism of human visual pathway and is the optimal model for building visual pathway replicated hardware. Secondly, the model mechanism is simple and easy to build through hardware circuits without adding too much cost and design difficulty. Finally, the model is effective and has the ability to improve or reconstruct the object motion tracking by optimizing the delay module design method.

Fig. R3 | Barlow-levick model¹. The Barlow-Levick detector calculates the direction of image velocity by processing the brightness values at two adjacent image points through a logical AND-NOT gate after one of them is delayed by ϵ ms.

References:

1. Borst, A. & Euler, T. Seeing Things in Motion: Models, Circuits, and Mechanisms. *Neuron* **71**,

974-994 (2011).

2. Strauss, S. et al. Center-surround interactions underlie bipolar cell motion sensitivity in the mouse retina. *Nat. Commun.* **13**, 1–18 (2022).

Response to Reviewer #4

General comments:

“I co-reviewed this manuscript with one of the reviewers who provided the listed reports. This is part of the Nature Communications initiative to facilitate training in peer review and to provide appropriate recognition for Early Career Researchers who co-review manuscripts.”

Response:

Thank you very much for your valuable comments, and hope the replies can be approved.

Response to REVIEWER COMMENTS

Response to Reviewer #1

General comments:

“The authors addressed most of my concerns during the first round of review, but some descriptions in the manuscript are still unclear, which may affect the readability of the manuscript.”

Response:

Thank you very much for your recognition and support. We add more descriptions in detail about your valuable concerns and hope to meet your expectations.

Comment 1:

“In the color processing section, Fig. 2a shows the distributions of red, green, and blue colors at different spatial locations. I have a few questions: Is the convolution of these three colors with the responsivity convolution kernel performed independently or simultaneously? Additionally, does the responsivity convolution kernel change depending on the color being processed? There is also an inconsistency between the convolution kernel values mentioned in the main text and the supplementary information, which needs clarification.”

Response:

Thank you very much for your comments!

The convolution of these three colors with the responsivity convolution kernel is performed independently, because of single-color input of the testing facility (Fig. S12).

The responsivity convolution kernel does not change depending on the color being processed. For large/small receptive fields, the responsivity convolution kernel has different absolute values according to the difference in Gaussian standard deviation σ . For different color, such as R+G⁻ and G⁻R⁺ of the same receptive field, the sign of the responsivity value at each site is determined by "+" and "-" without changing the absolute value (Fig. S13).

The convolution kernel values mentioned in the main text and the supplementary

information have the same absolute values. The description “the responsivity distribution of small (3×3) and large (5×5) CSRF with two values (0.04 and -0.008 A/W) and three values (0.007, 0.003 and -0.0001 A/W)” in the main text is just the example of small $R+G^-$ and large $G+R^-$. There are other circumstances: small $G-R+ / Y+B- / B-Y+$ / brightness and large $R-G+ / B+Y- / Y-B+$ / darkness in Fig. S13. For SO signal, small $R+G^-$ and $G-R+$ are added to get the $R+G^-$ SO signal. [“ $R+G^-$ ” + “ $G-R+$ ”] is equal to [full “ $R+$ ” + full “ $G-$ ”]. Therefore, we choose this equivalent test scenario under the limited single-color input of the testing facility (Fig. S12).

The relevant explanations are listed in the Supplementary Information:

[In each site, the responsivity value for R/G/B in the 3×3 or 5×5 CSRF is the same. The sign “+” / “-” in “ $R+$, $R-$, $G+$, $G-$, $B+$, $B-$ ” represents potentiated/ depressed information at the corresponding site.] (SI, line 133–135)

[In fact, the SO signal is get by adding the “ $R+G^-$ ” and “ $G-R+$ ”, which is equal to add full “ $R+$ ” and full “ $G-$ ”. Therefore, the convolution kernels for “ $R+$ ” and “ $G-$ ” are respectively set to all positive and negative values, because of single-color input of the testing facility.] (SI, line 147–150)

[Among them, six 3×3 and 5×5 retina-sensing modules carry out photoelectric conversion of small and large receptive fields R_+ , R_- , G_+ , G_- , B_+ , B_- , respectively. Limited by single-color input of the testing facility, the retina-sensing module is reused to record twelve voltage signals which are imported into subsequent circuits through a signal generator in experiment.] (SI, line 190–195)

Comment 2:

“Regarding the shape recognition demonstration, the reviewer remains unclear about how the demonstration was conducted. Specifically, for “The light-mask input is generated by illuminating the light mask of a right-angle triangle.”, why does illuminating a right-angle triangle mask generate a disordered light-mask input? Was this light-mask input randomly selected from a database of 1,600 samples? What was the basis for selecting the five OCKs shown in Fig. 3c, including two at 0° , two at 90° , and one at 45° ? Is the discrete neural network processing applied only to the five parts

within the red dashed box of the light-mask input? If so, why was a 15×15 pixels size light-mask input necessary? What is the purpose of the other light mask patterns shown in Fig. S1? I request the authors to provide more detailed explanations for each demonstration, as this would help readers better understand the application potentials of this retina-mimicked hardware.”

Response:

We are grateful for your valuable comments.

The disordered light-mask input is generated by the light mask consisting of a right-angle triangle mask and the frosted glass. The blocking metal of the mask is thin with certain light transmittance, and different Gaussian blur-like illumination inputs can be provided by moving the position of the glass.

The light-mask input is randomly selected from the database, but cannot be completely consistent with the database of 1,600 samples (Fig. S15a). The database is added with Gaussian noise only used for the training of neural network weights in simulation. The Gaussian blur-like illumination input generated by the light mask is used for the actual experimental measurement. The recognition rate of each epoch in experiment (Fig. 3f) is obtained by counting the results under 480 (30% of 1,600) Gaussian blur-like illumination inputs by moving the glass.

The selected five OCKs shown in Fig. 3c are based on the possible contours in the five specific sites of these four right-angle triangles. This is consistent with the way the visual pathway perceives shape information¹.

The human visual system recognizes the shape of objects, which is a discrete form based on the contour information of specific points. Therefore, only five parts within the red dashed box in Fig. 3c is processed.

The 15×15 pixels size light-mask is necessary for defining different types of right-angle triangles. Although only partial features are focused, identifying the object itself is the ultimate goal.

The other light mask patterns shown in Fig. S1 are used for other demonstrations. The left one is used for color processing. The middle regular hexadecagon mask is used for testing OCK's orientation selection effect. The light mask of the pattern “hust” is

used for testing motion tracking.

The relevant explanations are listed in the revised manuscript and Supplementary Information:

“In the shape recognition section, the mask consists of a mask and the frosted glass to generate disordered right-angle triangle light input.” (main text, line 328–330)

“The Gaussian blur-like illumination input generated by the light mask consisting of a right-angle triangle mask and the frosted glass is used for the actual experimental measurement. The recognition rate of each epoch in experiment (Fig. 3f) is obtained by counting the results under 480 (30% of 1,600) different Gaussian blur-like illumination inputs by moving the glass.” (SI, line 314–318)

“The left one is used for color processing. The regular hexadecagon of 50×50 pixels (middle) and four 15×15 -pixel size right-angled triangle (right) ones are used for shape recognition.” (SI, line 369–371)

References:

1. Brincat, S. L. & Connor, C. E. Underlying principles of visual shape selectivity in posterior inferotemporal cortex. *Nat. Neurosci.* **7**, 880–886 (2004).

Response to Reviewer #2

General comments:

“The revised version is almost ready for publication.”

Response:

Thank you very much for your recognition of this work.

Response to Reviewer #3

General comments:

“The authors have well addressed my comments in the revised manuscript, I have no further question on this manuscript and I am happy to recommend this great work for publication in Nature Communications.”

Response:

Thank you very much for your recognition and affirmation.

Response to Reviewer #4

General comments:

“I co-reviewed this manuscript with one of the reviewers who provided the listed reports. This is part of the Nature Communications initiative to facilitate training in peer review and to provide appropriate recognition for Early Career Researchers who co-review manuscripts.”

Response:

Thank you very much for your valuable comments, and hope the replies can be approved.